# HCMV Antivirals and Strategies to Target the Latent Reservoir

**DOI:** 10.3390/v13050817

**Published:** 2021-05-01

**Authors:** Marianne R. Perera, Mark R. Wills, John H. Sinclair

**Affiliations:** Department of Medicine, Cambridge Institute of Therapeutic Immunology and Infectious Disease, University of Cambridge, Addenbrooke’s Hospital, Hills Road, Cambridge CB2 0QQ, UK; mp704@cam.ac.uk (M.R.P.); mrw1004@cam.ac.uk (M.R.W.)

**Keywords:** human cytomegalovirus, latency, antiviral, latent reservoir, shock and kill, F49A-FTP, transplant

## Abstract

Human cytomegalovirus (HCMV) is a ubiquitous human herpesvirus. In healthy people, primary infection is generally asymptomatic, and the virus can go on to establish lifelong latency in cells of the myeloid lineage. However, HCMV often causes severe disease in the immunosuppressed: transplant recipients and people living with AIDS, and also in the immunonaive foetus. At present, there are several antiviral drugs licensed to control HCMV disease. However, these are all faced with problems of poor bioavailability, toxicity and rapidly emerging viral resistance. Furthermore, none of them are capable of fully clearing the virus from the host, as they do not target latent infection. Consequently, reactivation from latency is a significant source of disease, and there remains an unmet need for treatments that also target latent infection. This review briefly summarises the most common HCMV antivirals used in clinic at present and discusses current research into targeting the latent HCMV reservoir.

## 1. Introduction

Human cytomegalovirus (HCMV) is a betaherpesvirus that is a remarkably successful pathogen, infecting approximately 40 to 100% of the population worldwide [1]. Primary infection of healthy individuals with HCMV is typically asymptomatic, although it can present with mild flu-like symptoms. The virus replicates in a broad range of cell types; briefly, after entry via fusion or receptor-mediated endocytosis, the viral genome is delivered to the nucleus where it expresses a cascade of temporally regulated genes [2]. This is followed by capsid assembly, the acquisition of tegument proteins, envelopment in a lipid bilayer and egress of the new virions (reviewed in [2]). A rapid and robust antiviral response is mounted against HCMV, in which innate, cellular and humoral immunity play an important role [3,4,5]. However, despite this, the virus is never cleared and goes on to establish a latent infection in myeloid progenitor cells in the bone marrow, where it can reside for the remainder of the host’s lifetime [6,7,8]. It does, however, periodically reactivate, but these reactivation events are quickly shut down by the immune system, generally resulting in a sub-clinical infection [5].

Problematically, when the immune system is unable to control either primary infection, reinfection with a new virus strain or reactivation of virus from latency, HCMV replicates and disseminates, often causing disease [9]. This is routinely observed in the immuno-incompetent: immunosuppressed transplant recipients, patients living with AIDS and foetuses (congenital CMV), and has severe implications for their health.

In the UK, congenital HCMV infection occurs in approximately 0.8% of live births [10]. Around 10–15% of congenitally infected infants show symptoms at birth, including splenomegaly, hepatomegaly, thrombocytopenia, microcephaly and sensorineural hearing loss [10]. Of those infected infants that are asymptomatic at birth, around 15% go on to develop complications, including sensorineural hearing loss and mental retardation [10]. Congenital HCMV infection is the leading cause of non-genetic hearing loss [11].

Approximately 2% of seronegative pregnant women seroconvert by the time they give birth [9]. In pregnancy, the virus can cross the placenta to cause intrauterine infections, 20% of which go on to manifest as severe disease in the foetus [12,13]. Not only primary infection, but also reinfection with a new strain, or reactivation from latency in seropositive women, can have serious consequences for the foetus [13,14,15]. Roughly 1% of women who have already been infected with HCMV before their pregnancy give birth to children with congenital HCMV infection [16,17,18]. In fact, although transmission to the foetus is much more efficient following primary infection, the high seroprevalence of HCMV means that, each year, mothers who are already seropositive actually deliver more congenitally affected children than mothers who undergo a primary infection during pregnancy [9,13,14].

As an iatrogenic pathogen, HCMV also causes significant morbidity in the transplant setting. Under the immunosuppressive treatments that accompany transplantation, patients are no longer able to control HCMV infection. Since the virus has an extremely broad tropism for different human tissues, it can lead to severe disease in multiple organs, including hepatitis, gastrointestinal disease and pneumonia [19]. Lysis of cells during its replication cycle, apoptosis and necrosis of infected cells, and secondary damage by the immune response can all cause extensive damage to host tissue [20].

In solid organ transplants (SOT), approximately 78% of seropositive donors (D+) transmit HCMV to seronegative recipients (R−) in allografts containing lytic or latent virus [21,22]. This combination of D+/R− in SOTs poses the highest risk of severe CMV disease, exacerbated by the lack of any pre-existing immunity of the recipient. However, akin to congenital CMV, prior exposure to the virus does not preclude disease; around 40% of seropositive SOT recipients (R+) see reactivation of their own latent reservoirs of HCMV during immunosuppressive treatment, and some are also reinfected by new HCMV strains transmitted by a seropositive donor [21,23].

By contrast, in bone marrow and peripheral haematopoietic stem cell transplantation (HSCT), the most at risk group for CMV disease is seropositive recipients with a seronegative donor (D−/R+) [24,25,26]. In these patients, their own latent virus reactivates, but the donor graft provides no HCMV antigen-specific T cells to control its spread.

Although it is not associated with severe disease in immunocompetent people, there have also been reports that HCMV may contribute to long-term morbidity in otherwise healthy individuals by promoting diseases such as cancer and atherosclerosis [27,28].

Efforts into making a vaccine against HCMV began in the 1970s, but as of yet, no vaccine is licensed for use [29]. There are currently several approved drugs available for HCMV disease, which will be briefly summarised below.

## 2. Current HCMV Antivirals

### 2.1. Ganciclovir

Ganciclovir, GCV, and its valine ester derivative, valganciclovir, are first-line drugs used against CMV disease. GCV is an acyclic nucleoside analogue of deoxyguanosine [30]. It is phosphorylated by an HCMV kinase, pUL97, to produce ganciclovir monophosphate [31,32]. The charge gained by phosphorylation prevents diffusion of the drug back out the cell, allowing high concentrations to build up in infected cells [33]. It can then be further phosphorylated by cellular kinases to ganciclovir di- and then triphosphate, which competes with dGTP for the active site of the HCMV DNA polymerase, pUL54, and acts as a chain terminator during viral DNA replication [34,35].

Unlike many other chain terminators, ganciclovir does have an equivalent to a 3′ hydroxyl group, which can be used to append one further base. However, the addition of ganciclovir to the nascent DNA causes rapid excision of nucleotides two bases downstream of GCV, preventing any further elongation [36].

GCV’s effects are highly specific to infected cells: as well as its requirement for activation by a viral kinase, GCV also inhibits the viral DNA polymerase much more effectively than the cellular DNA polymerase [37].

Oral absorption of GCV is poor, but improved by valine esterification of the drug to valganciclovir, which is then cleaved to ganciclovir in the intestines or liver [38,39]. Adverse effects of ganciclovir include neutropenia, thrombocytopenia and nephrotoxicity [40]. Resistance to GCV arises mainly via mutations in UL97, which prevent efficient GCV phosphorylation, and/or in the viral polymerase, UL54 [41].

GCV is administered to patients at risk of CMV disease after solid organ transplant or haematopoietic stem cell transplant. Two main strategies exist for limiting CMV disease in these patients: either universal prophylaxis or pre-emptive therapy [42]. In universal prophylaxis, all at-risk patients receive GCV treatment for 100 days post-transplant. Treatment is limited to 100 days to minimize toxic side effects [43]. Pre-emptive therapy involves regularly monitoring patients for signs of CMV replication (viral DNA or antigens in the blood) and starting GCV treatment if these are detected. Pre-emptive therapy is preferred for HSCT patients in the UK; the myelosuppressive side effects of GCV can interfere with immune reconstitution and leave these patients susceptible to bacterial and fungal infections [44,45].

### 2.2. Foscarnet

Foscarnet and cidofovir are administered as second-line drugs in the case of resistance to ganciclovir. They do not require activation by viral pUL97, and so are suitable for use against GCV-resistant UL97 mutants.

Foscarnet is a structural analogue to pyrophosphate (PPi). During the elongation step of DNA replication, pyrophosphate is removed from dNTPs before their incorporation into the nascent strand. Foscarnet acts as a reversible product inhibitor, binding to the HCMV DNA polymerase (UL54) binding site for pyrophosphate, and preventing PPi from being cleaved from incoming dNTPs [46,47]. Foscarnet has a 100-fold higher affinity for viral DNA polymerase over cellular DNA polymerase [48]. Like GCV, a major side effect of foscarnet is nephrotoxicity, although it seems to cause minimal myelosuppression [49].

### 2.3. Cidofovir

Cidofovir is a structural analogue of cytosine monophosphate. Since it already has a single phosphate, it does not require initial activation by UL97, in contrast to ganciclovir [50]. It is phosphorylated by cellular kinases to its active form. Incorporation of cidofovir into the nascent DNA chain drastically slows down the rate of elongation. If two consecutive cidofovirs are appended, the chain cannot grow any further [51].

The main side effects of cidofovir are nephrotoxicity and myelosuppression [52]. An additional problem with cidofovir is poor bioavailability; hence a modified lipid conjugate of cidofovir, brincidofovir, which has improved bioavalability, is currently undergoing clinical trials for efficacy against HCMV disease [53]. Brincidofovir displays lower nephrotoxicity because, unlike cidofovir, it cannot be imported by organic anion transporter 1 into epithelial cells of the renal proximal tubule [54].

### 2.4. Letermovir

Although foscarnet and cidofovir do not require activation by UL97, they both target UL54, so cross-resistance after ganciclovir treatment can still occur [55]. A more recently FDA-approved anti-HCMV drug, letermovir, avoids this issue by inhibiting a completely different target. Letermovir inhibits pUL56, a component of the terminase complex (a trimer of pUL51, pUL56 and pUL89), which packages the viral genome into the capsid [56]. No homologue of pUL56 has been identified in mammalian cells, allowing the drug to target the virus selectively. Unfortunately, resistance to letermovir has also been reported, which can be traced back to mutations in UL56 [57]. Adverse gastrointestinal effects have been reported with letermovir, but overall, it appears to cause fewer side effects than other approved HCMV antivirals, which has been attributed to the lack of any mammalian counterpart to UL56 [58,59].

### 2.5. Maribavir

Although not yet licensed for use, the prospective anti-HCMV drug, maribavir, has had promising results in clinical trials [60]. Maribavir competes with ATP for binding to the viral kinase pUL97. Maribavir is still active against some GCV-resistant strains of HCMV, despite the majority of GCV-resistant strains having mutations in UL97 [61]. It should be noted that maribavir could not be used in conjunction with GCV treatment as it would inhibit the initial phosphorylation and activation step of GCV [62].

### 2.6. Antibody Therapies

A vaccine against HCMV is considered to be a very high priority, particularly for the prevention of congenital disease, but none has been licensed [63]. Due to the problems described above with respect to current antivirals, as one alternative, antibody responses have been investigated as a basis for immunotherapies [64]. For instance, hyperimmune globulin (HIG) can improve survival in patients undergoing solid organ transplantation and a range of antibodies that are capable of neutralizing the entry of cell-free virus have been developed [65,66]. However, so far, the use of neutralizing therapeutic monoclonal antibodies has only had modest effects in trials [67].

## 3. Strategies for Targeting the Latent HCMV Reservoir

Aside from issues with resistance, poor bioavailability and toxicity in already sick patients, current HCMV antivirals only stop lytic virus replication and cannot eliminate latent reservoirs of virus [25]. Therefore, HCMV is never cleared from the host and after antiviral treatment is ceased, virus replication is no longer inhibited and CMV disease has the potential to recur [43]. As described in the introduction, reactivation of virus from latency is a significant source of disease in both congenital CMV and the transplant setting. There is also some evidence that long-term infection with HCMV contributes to chronic illnesses like atherosclerosis and cancers in the immunocompetent [27].

Latency of HCMV is defined as the carriage of viral genomes in the absence of the production of infectious virus. HCMV can undergo latent infection in cells of the early myeloid lineage, such as CD34+ haematopoietic progenitor cells and their derivative CD14+ monocytes [6,7,8,63,64]. An open question is whether HCMV latency occurs in other sites besides the myeloid lineage in vivo. Endothelial cells have been suggested to be another important source of latent virus in SOT [65] and although one study did not identify latently infected endothelial cells in the saphenous veins of seropositive subjects, endothelial cells are highly heterogeneous [68,69,70]. For instance, sinusoidal endothelial cells and peritubular capillary endothelial cells have been identified as sites of murine cytomegalovirus (MCMV) latency in the liver and kidney but these sites have not been investigated as potential sites of HCMV latency in human organs [71,72].

In cells of the early myeloid lineage, the latent virus has a much more restricted gene expression profile, expressing only a subset of those genes seen in a lytic infection, although this may be much wider than initially thought [73,74,75,76,77]. It is also not known whether different sites of myeloid latency (e.g., CD34+ progenitor cells and their derivative CD14+ monocytes) have different latency-associated gene expression signatures. Crucially, in both cell types, the major immediate early promoter/enhancer (MIEP) remains suppressed during latency [78,79]. During latency, viral DNA replication does not occur, and many viral antigens are not expressed. Therefore, as well as escaping immune recognition, latently infected cells are also safe from current HCMV antivirals, which also only target lytic replication. However, this latent reservoir of virus in undifferentiated myeloid cells can reactivate during myeloid cell differentiation and/or if the latent cell is subject to inflammatory environments [80].

One option to avoid HCMV disease derived from a host’s latent reservoirs would be to inhibit reactivation of the virus and keep the latent reservoir suppressed in the immune-incompetent, a so-called ‘block and lock’ approach [81,82]. Interferon β has been found to prevent MCMV reactivation in naturally latent cells [83,84]. Drugs that inhibit HCMV reactivation have also been identified. For example, histone acetyl transferases (HATs) are known to be important in the de-repression of the MIEP and reactivation of HCMV [85]. Inhibition of HATs with the drug MG149 resulted in a decrease in reactivation of latent HCMV [85]. Additionally, UL33 was recently found to be important for reactivation of HCMV via phosphorylation of CREB. Pharmacological inhibition of CREB phosphorylation with 666-15 reduced reactivation frequencies without having a significant impact on cell viability [86,87]. However, reactivation inhibitors might have to be taken indefinitely to keep the virus suppressed which is not an effective solution, especially for congenital CMV, where foetus development might be affected by epigenetic modulators. The ideal therapy would be a single course of treatment that purges the latent reservoir.

Another approach to target the latent reservoir would be to temporarily drive the virus out of latency so latently infected cells transiently express lytic viral antigens. The immune system would then be able to recognise and clear these otherwise latently infected cells. This strategy, termed ‘shock and kill’, is also being widely investigated as a therapeutic tool to target HIV-1 latency [88,89]. In essence, epigenetic modifiers have been used to reactivate HIV-1 gene expression in latently infected cells to allow them to be targeted by pre-existing or newly induced HIV-1 specific immune responses. Similar shock and kill strategies against HCMV are discussed in the next section. Alternatively, biological properties of essential viral genes expressed during latency and/or identification of changes in the latently infected cell might allow it to be directly targeted, and these are discussed after shock and kill strategies below.

### 3.1. Potential ‘Shock and Kill’ Treatments to Purge the Latent HCMV Reservoir

HCMV latency is, in part, maintained via silencing of the MIEP, which prevents high levels of expression of the viral major immediate early (IE) products which are required for virus lytic replication. During latent infection in undifferentiated myeloid cells, the MIEP becomes associated with markers of repressive chromatin, including deacetylated/methylated histones [90,91,92]. By contrast, upon reactivation, these repressive chromatin marks are removed and replaced with chromatin markers of active transcription, such as acetylated histones, resulting in de-repression of the MIEP and the initiation of lytic HCMV infection [93].

A combination of viral and cellular factors are known to act to modulate the transcriptional activity of the MIE locus [94,95,96], for reviews see [79,97,98]. One such cellular factor is the histone deacetylase HDAC4, which is upregulated in latently infected cells and associates with the MIEP, deacetylating histones to aid in its repression [99]. Therefore, some time ago, it was hypothesized that HDAC inhibitors (HDACis) should be able to reverse histone deacetylation at the MIEP causing its de-repression and the reactivation of lytic major IE gene expression in these otherwise latently infected cells (the so-called ‘shock’) [99]. These would then be targets for HCMV-specific cytotoxic T cells (‘kill’). One biological property of HCMV that makes this type of strategy so attractive is the knowledge that in healthy seropositive virus carriers, up to 10% of circulating peripheral blood resident cytotoxic T lymphocytes (CTLs) are HCMV lytic antigen specific and potently target lytically infected cells [100].

As anticipated, treatment of both experimentally and naturally latently infected monocytes with an HDAC4 inhibitor (MC1568) did result in de-repression of the MIEP and the transient induction of expression of lytic IE antigens, though the full complement of viral lytic genes was not induced by MC1568 treatment [99]. This could be construed as the most desirable outcome for a latency reversing agent for HCMV as the treatment does not result in full reactivation of the lytic viral life cycle, thereby reducing the possibility of transplanting infectious virus. Instead, latently infected cells induced to transiently express, e.g., IE antigens would now become a viable target for the host immune system; an extraordinary proportion (between 0.1 and 5%) of total circulating CD8+ CTLs in seropositive donors respond to peptides derived from IE72 [101].

Indeed, when naturally latent monocytes from a seropositive donor were treated with MC1568, the donor’s autologous T cells were able to recognise and clear them, resulting in a substantial reduction in the number of cells capable of reactivating virus after differentiation [99]. However, these analyses were somewhat compromised by the fact that treatment with MC1568 was carried out transiently; purified latently infected monocytes were treated for 48 h with MC1568 in the absence of other peripheral blood mononuclear cells which would be unlikely to lend itself to any treatment regimen in vivo.

With this in mind, Groves et al. (2021) built on this work and screened a new generation of HDACis that had lower toxicity and resulted in a substantially higher number of monocytes expressing lytic HCMV antigens in a model of a long-term treatment regime of latently infected monocytes [102]. However, in these long-term treatment studies, both old and new HDACis also resulted in full reactivation of infectious virus. Consequently, the authors also trialled inhibitors of other classes of epigenetic enzymes, such as inhibitors of BET proteins (I-BETs) for their potential as ‘latency reversing agents’, with far more success [103].

At a large number of metazoan promoters, RNA Pol II remains ‘paused’ due to the presence of negative transcription elongation factors such as NELF and DSIF [104]. The transcription factor P-TEFb (comprising CDK9 and Cyclin T1) can overcome this promoter pausing and allow elongation of the nascent RNA transcript to continue. However, PTEFb may be bound to the inactivating 7SK snRNP or complexed to the BET protein, Brd4, reducing levels of free PTEFb. Treatment with I-BETs can release more PTEFb, resulting in increased transcription from Brd4-independent genes [105,106].

In the study by Groves et al. (2021), an I-BET, GSK726 was far superior to HDACis in its ability to induce IE gene expression in latently infected CD14+ monocytes [102]. Furthermore, unlike most new generation HDACis tested, GSK726 did not result in full reactivation of the virus, but the expression of a select group of lytic antigens, even after long-term incubation. This included the immediate early proteins, IE72 and IE86, as well as the late protein, pp65, but not the early protein UL44. Importantly, treatment with the BET inhibitor did not result in viral DNA replication. HCMV lytic gene expression normally occurs in a tightly controlled, orderly cascade of immediate early (IE), then early (E), followed by DNA replication and late (L) gene expression. RNAseq confirmed that GSK726, unliked the HDACis, was activating a certain set of viral promoters rather than initiating the lytic life cycle [102].

As well as causing IE72 and IE86 expression, I-BET treatment also drove the expression of other immunodominant HCMV antigens which are major targets for the adaptive immune response, including gB and pp65. One study found that between 40 and 70% of the neutralizing antibody response from seropositive individuals was specific to a single viral envelope glycoprotein, gB, which is the most abundant envelope protein [107]. The tegument protein, pp65 has been found to be specifically targeted by 70–90% of HCMV-specific cytotoxic T lymphocytes (CTLs) [108].

Furthermore, despite derepressing a range of lytic genes, HCMV encoded immunoevasins were not part of the select group of viral proteins induced by I-BET treatment. I-BET treatment actually resulted in a decrease in the transcription of viral genes with roles in interfering with antigen presentation and T cell recognition: UL21.5, US8, UL18, US10, UL6-11, US2, US3, US6, and US11. Genes which were not activated by I-BET treatment likely depended on Brd4 activity for expression; at certain promoters, Brd4 is actually required to recruit PTEFb [102]. Additionally, it was found that I-BETs actually inhibited viral DNA replication in permissive fibroblasts [102].

One of the other limitations of HDACis as a ‘shock and kill’ treatment is the fact they can sometimes impair cytotoxic T cell function. However, assessment of the effect of I-BETs on T cell activity against a range of HCMV peptides showed that I-BETs had no detrimental effect. Consistent with this, the addition of autologous T cells to either experimentally infected or naturally latent monocytes in the long-term presence of BET inhibitors (more equivalent to any in vivo treatment strategy with I-BETs) resulted in the clearance of latently infected cells [102]. I-BETs, therefore, appear to strike an optimal balance: derepressing immunodominant antigens whilst keeping immunoevasins silenced, and even actively inhibiting viral DNA replication [102]. They showed no adverse effects on T cell effector function or cell viability, and so they are being examined further as a possible treatment for latent HCMV reservoirs in vivo [102].

Other hypothetical drugs to shock and kill latent HCMV could also be explored. For example, it has been shown that treatment with TNFα leads to reactivation of HCMV from latency via NFκB signalling in a cell line model of CD34+ progenitor cells [109]. Furthermore, TNFR1 is upregulated on latently infected cells by the viral protein pUL138, which could potentially make these cells more responsive to TNFα treatment over uninfected bystander cells [110]. However, extreme caution would have to be applied in stimulating immune cells (in the case of HSCT) with the pro-inflammatory cytokine TNFα. It could exacerbate graft vs. host disease and also hamper haematopoietic reconstitution [111,112]. Others have shown that MEK/ERK and PI3K/AKT signalling is important for establishing and maintaining latency [86,113,114]. Pharmacological inhibition of MEK, ERK, PI3K or AKT has been shown to result in HCMV reactivation [115]. These drugs could therefore be examined for their potential as shock and kill agents. Furthermore, Rauwel et al. showed that phosphorylation of the cellular protein, KAP1, prevented recruitment of repressors SETDB1 and HP1 to the MIEP during latency [96]. Drug-induced phosphorylation of KAP1 by the ATM activator, chloroquine, led to the reactivation of HCMV in naturally latent monocytes [96].

One downside, however, to all the treatments discussed above is that there might be broad off-target effects since, e.g., HDACis and I-BETs target important cellular functions; namely host chromatin modification and transcriptional control. Whilst any in vivo strategy would likely be restricted to short-term treatment to purge latently infected cells, it would also be advantageous to target specifically the latently infected cells to reduce side effects, for example by targeting a virally encoded gene expressed during latency.

### 3.2. Eradicating the Latent Reservoir by Targeting Viral Genes Expressed during Latency

Although the quality and quantity of viral gene expression is altered and reduced during latent carriage, a number of viral genes are expressed to significant levels during latency [77,91]. The most highly transcribed of these are non-immunogenic long non-coding RNAs [77,91]. However, the latent virus also expresses protein coding genes which should theoretically be targetable by the immune system, but these antigens still manage to avoid detection via a number of latency-associated immune evasion strategies employed by HCMV [26]. Nevertheless, these viral proteins could still be potential targets for drugs.

#### 3.2.1. Vincristine and UL138

The viral protein UL138, a membrane-localised protein, is an important latency-associated viral gene product [95,116]. Whilst there is currently no direct way to target UL138 specifically, the consequences of UL138 expression can be exploited as an “Achilles heel” for latently infected cells. UL138 expression during latency is believed to result in a number of changes in cell phenotype to optimise the cellular milieu for latent HCMV infection [97]. One such change is the UL138-mediated downregulation of cellular MRP-1. MRP-1 is a cell surface protein that exports vinca alkaloids such as doxorubicin and vincristine and is known to contribute to tumour resistance [117].

On this basis, Weekes et al. (2013) tested whether vincristine, which inhibits tubule polymerisation and results in cell death (and is normally efficiently exported by MRP-1), would disproportionately affect latently infected cells because of their UL138-mediated down-regulation of MRP-1 [118]. The authors found vincristine did, indeed, target and kill experimentally and naturally latently infected monocytes as well as naturally latently infected CD34+ progenitor cells [118]. Unfortunately, though, the inherent toxicity of vincristine would likely limit its use as a treatment to target the latent HCMV reservoir in vivo. Nevertheless, this proved an important proof of principal that differences in latently infected cells could make them targetable by novel therapeutic strategies [119].

#### 3.2.2. F49A Fusion Toxin Protein to Target US28

Another viral protein whose expression is essential for latent carriage of HCMV is US28, a viral G-protein coupled receptor (GPCR) and chemokine receptor homologue which is expressed on the surface of latently infected monocytes and binds the CX3CL1 ligand more strongly than the endogenous receptor, CX3CR1 [120,121]. Natural antibodies to US28 in seropositive individuals have been shown to target latently infected monocytes for killing by neutrophils; however, latently infected monocytes downregulate secretion of neutrophil chemoattractant proteins S100A8/A9, resulting in avoidance of neutrophil recruitment by the latently infected cells [122].

US28 expression during latent carriage has recently been used as another novel way of targeting the latent reservoir: utilising a US28-specific fusion toxin protein (F49A-FTP) to target and kill latently infected cells [120]. Fusion toxin proteins can be used to target specific proteins present on the surface of cells. They are composed of cellular toxins joined to a targeting domain which is often the Fab fragment of an antibody, but can also be a ligand for a receptor. F49A-FTP comprises a Pseudomonas exotoxin domain fused to CX3CL1, which has itself been mutated to bind to US28 with an even higher affinity [123]. After binding to US28, F49A-FTP is internalised and its exotoxin domain ADP-ribosylates elongation factor-2 (EF-2), which inhibits translation in the target cell and results in cell death.

Krishna et al. (2017) showed that treatment with F49A-FTP killed latently infected monocytes, resulting in a substantial decrease in levels of reactivatable virus [120]. This is consistent with the fusion toxin protein selectively binding and killing cells in the latent reservoir [120]. F49A-FTP also successfully targeted latently infected CD34+ haematopoietic progenitor cells, another site of HCMV latency. Finally, F49A-FTP was also shown to work in the context of natural latency, reducing reactivation of the virus from naturally infected monocytes present in a seropositive donor.

#### 3.2.3. US28 Inhibitors as a Novel Method for ‘Shock and Kill’

An important function of US28 during latency is the suppression of MAPK and NF-κB signalling to repress MIEP activity and prevent lytic IE gene expression [113]. Consequently, it has been posited that, besides killing US28 expressing cells directly with a US28-specific fusion toxin protein, an alternative therapeutic strategy would be to target US28’s MIEP repressive function to induce IE expression [113]. This could be employed as a shock and kill strategy: inhibition of US28 should have a similar effect to HDACis and I-BETs by forcing inappropriate expression of lytic antigens in otherwise latently infected cells and allowing their subsequent recognition by HCMV-specific CTLs. Importantly, this ‘shock and kill’ approach would likely have none of the drawbacks of inhibiting a cellular target like HDACs and BET proteins; a viral target should allow greater specificity and fewer off-target effects.

With this in mind, Krishna et al. (2017) employed an inverse agonist of US28 (VUF2274) which binds US28 and inhibits US28’s downstream MIEP suppressive signalling cascade. Treatment of latently infected monocytes with varying concentrations of VUF2274 resulted in the induction of IE expression in these latently infected cells, resulting in their targeting and killing by IE-specific CTLs. Unfortunately, VUF2274 displayed significant toxicity towards uninfected monocytes, possibly in part because it is also an antagonist for cellular chemokine receptor, CCR1, which has 30% homology to US28 [113,124]. This makes it less appropriate as a ‘shock and kill’ agent. However, novel strategies are being developed to inhibit US28 more selectively and drive IE expression for other shock and kill approaches [125].

## 4. Targeting the Latent Reservoir in the Transplant Scenario

Clearly, the ability to target the latent reservoir to directly kill latently infected cells or to induce transient IE activation as the basis for shock and kill strategies is becoming a serious possibility.

These treatments could either be administered to the live donor, prior to graft removal, or applied just to the tissue being transplanted: by perfusing an organ or treating HSCT transplant cells in vitro [26]. In cases where a transplant recipient is also seropositive, these strategies could be used to clear their existing viral reservoirs prior to engraftment and immunosuppression (see Figure 1a,b). The possibility of targeting viral pathogens in donor organs prior to engraftment is not without precedent. Ex vivo delivery of monoclonal antibody to target EBV as well as light-based therapies to target HCV in donor lungs prior to transplant have been described [126,127]. Similarly, removal of macrophages from RCMV-infected donor hearts prior to transplant reduces chronic rejection [128].

F49A-FTP has already been trialled in an ex vivo lung perfusion (EVLP) study with very promising results [129]. A small but increasing number of lung transplants are subject to EVLP, where they are kept on ventilators in organ care systems at 37 °C whilst a perfusate is pumped through the lung vasculature. This maintains the lung in more physiological conditions than the usual storage on ice, allowing time for evaluation of lungs that were initially deemed too risky to transplant [130,131]. Apart from increasing the number of lungs available for transplant, EVLP also provides an opportunity to treat the organ with drugs or other therapeutic agents before transplantation into the recipient [127,132].

In the study by Ribeiro et al., donor lungs rejected for transplant were perfused with or without F49A-FTP. Subsequently, sections of all the lungs were biopsied, and monocytes from these samples were extracted and differentiated in vitro to reactivate any remaining latent HCMV. The authors found a clear difference in the amount of recoverable HCMV; 100% of control samples reactivated virus, versus only 0.04% of F49A-FTP treated samples [129].

Since US28 shares 38% homology with cellular receptor, CX3CR1, the authors also tested for adverse effects on uninfected cells. After F49A-FTP treatment, no decline was found in the total number of CD14+ and CD34+ cells, which both express CX3CR1. Only a very small proportion of CD14+ and CD34+ cells are naturally latently infected, so a significant decrease in the total number of these cells would not be expected [133]. Additionally, they noted no difference in lung function between control and treated organs.

Normothermic perfusion of kidneys prior to transplant is also becoming a more common procedure [134] and similar proof of principle analyses are also being conducted in kidneys in ex vivo organ care systems prior to transplant (Hosgood and Nicholson, University of Cambridge, personal communication).

## 5. Conclusions

The prognosis of immunosuppressed transplant recipients with HCMV disease would be extremely poor without antiviral drugs, but even these treatments are not without their disadvantages: poor bioavailability, toxicity, emerging resistance and a failure to target and clear the latent reservoir, which remains an important source of disease. Our increased understanding of the basic molecular biology of latency and reactivation of HCMV has, however, now resulted in the real possibility of translating this knowledge into novel technologies to target the latent reservoir in the transplant setting.

One of the options being explored to target latently infected cells is ‘shock and kill’, where inhibiting a cellular or viral gene forces the virus to transiently express viral antigens which can be recognised and killed by existing HCMV-specific immune cells which are abundant in healthy seropositive individuals. Alternatively, although the immune system is oblivious to viral antigens expressed during latency, these could be targeted directly by, e.g., fusion toxin proteins, or phenotypic changes resulting from latent infection could be exploited to target the cell for killing.

Whilst targeting a cellular protein could lead to broad off-target effects, it is likely that such therapies would be brief and modified to minimise adverse effects, or they could be applied to the donor tissue/organ in isolation. Similarly, with respect to novel therapies targeting a viral gene, although the generation of viral resistance is a significant challenge for treating lytic HCMV infection, this would likely be less of a problem during latent infection due to the lack of viral DNA replication during latency.

Based on the initial promising results in a number of studies, further translational research is currently underway to determine if therapies targeting the latent HCMV reservoir would be suitable for use in clinic, particularly in the transplant scenario.

## Figures and Tables

**Figure 1 viruses-13-00817-f001:**
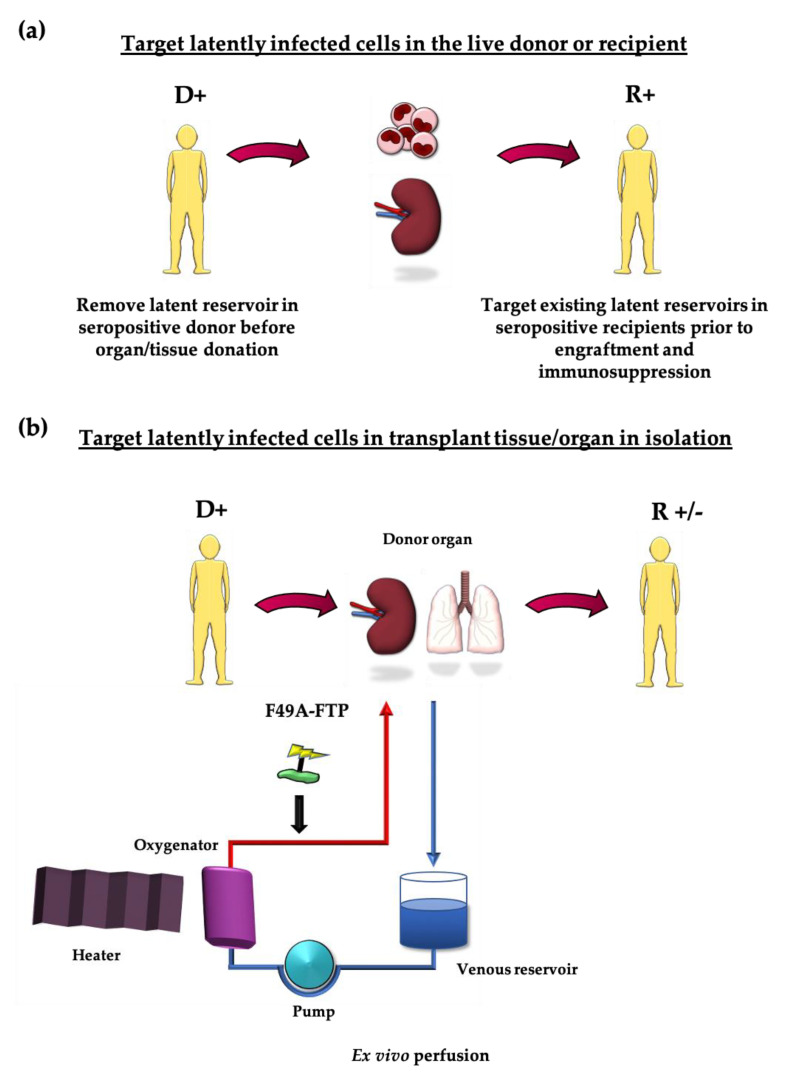
Opportunities for purging the latent reservoir (**a**) Seropositive donors could be treated before they donate organs/tissue to purge cells latently infected with HCMV. Seropositive recipients could likewise be treated prior to immunosuppression to remove their own existing latently infected cells that could reactivate. (**b**) Organs could be treated in isolation in ex vivo perfusion systems to remove latently infected cells from seropositive donors.

## Data Availability

The study does not report any primary data.

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
