# Peer review of "HCMV Antivirals and Strategies to Target the Latent Reservoir"

_viruses, 2021, doi:10.3390/v13050817_

Round 1
Reviewer 1 Report
The manuscript by Perera & Sinclair offers a comprehensive short overview about the current HCMV-specific antivirals in clinical use and gives an outlook on the recent status of strategies to target reservoirs of latent virus. The manuscript is well written and gives the non-in-field scientist a good choice to understand the mechanisms behind the presented strategies. Nevertheless, some minor points should be addressed before publication.
Specific points:
- The authors mainly discuss CD34+ stem and progenitor cells as the major site of HCMV latency e.g. lines 27 and 148 to 160. This is might be correct so far if most experimental systems are based on these cells and as in humans, HCMV can be transmitted by HSCT. But for SOT the setting of highest risk is D+/R-, here the chance of transfer of CD34+ cells is extremely low due to perfusion of the organ. Therefore, also other sites of HCMV latency are relevant e.g. endothelial cells and should be discussed. For a recent review also see Reddehase MJ, Lemmermann NAW. Cellular reservoirs of latent cytomegaloviruses. Med Microbiol Immunol. 2019 208(3-4):391-403. doi:10.1007/s00430-019-00592-y.
Further, it would be useful if the authors can discuss if there are differences in the latent pools (e.g. expression of viral markers, epigenetic regulation of transcription) that might influence the hit and kill strategy.
- Some references to review articles might be a little outdated, e.g. 2, 14, 16, 22, 23, and could be updated with newer ones.
- In paragraph 2.1 the important problem of pUL97 and pUL54 escape mutations should be mentioned.
- It would be great if the authors can also include observed side effects for Cidofovir and Letermovir to complete the section.
- For the more common reader, the paragraph 232 to 251 seems to be a little overwhelming
- Have the authors an explanation why after GSK726 treatment only some genes were enhanced in their transcription? Is there any information about cell-type specificity for the I-BETs?
- Comment to Figure 1a. For me, the lungs are a problematic organ for live donor donation.
Reviewer 2 Report
Perera and Sinclair's review on existing and new strategies to target latent HCMV is important and interesting. The author thoroughly describe several more recent studies on HDAC and I-BET inhibitors in detail, however the focus of the review is quite biased to these two approaches and lacks any discussion on several other approaches to the detriment of the overview theme proposed as the purpose of this review in both the abstract and introduction. I have attempted to present my comments in a topic-oriented approach below in order to make them more useful. In general, I believe, that with the addition of a broader literature representation, that this review would be useful and relevant to both CMV latency and the broader Viruses' audience.
1) Emphasis and audience:
- a small intro paragraph on CMV biology / replication kinetics would be more useful at the beginning (before section 2). For example, line 259 could move here.
- Line 178: the literature on HIV latency reversal is extensive, and the functional overlaps here are very relevant. It would be significant (and broader impact both) to include at least a few sentences on the key HIV studies/field here for perspective and broader context.
2) Missing factual references:
- lines: 57, 207, 285/287
3) Content not discussed or that needs included for balance:
- current treatments also include antibody therapy. Although the title suggests focus more on antivirals, specific targeted antibodies to US28 are included, so some mention of current antibody therapy is relevant.
- Section 2.3 is short and does not include the details as previous. For example what are the side effects, mechanism of action etc for letermovir (as described for the previous antivirals)?
- Section 2 also doesn't discuss the newer antivirals, i.e. Maribavir, Brincidofovir, Filociclovir. Or mAbs, or Sirtuins.
- Section 3: there are a number of other drugs that function to reactivate virus from other groups that should also be discussed. As two examples only (the others should also be included): PI3K/AKT/MEK/ERK inhibitors (Buehler, PLoS Pathogens, 2019), TNFa (Forte, mBio, 2018).
- Page 5 includes a lot of repeats from the discussion sections of the two main referenced papers here, the author's could summarize this section more succinctly and avoid the bias to these.
- Section 4: targeting during transplant. There is extensive work using HCMV homologs in animal models for transplant that should be acknowledged, i.e. MCMV renal transplant (Abecassis lab) and RCMV heart transplant (Streblow lab). There is also evidence for ex vivo treatments in other scenarios (outside if CMV) that could be referenced to provide proof of principle to this approach (i.e. Ku, 2020 for EBV; Galasso, Nat. Comm, 2019 lung transplant). -- If the desired focus of this review is to emphasize HDAC inhibitors for example rather than general latency reversal targets, this would be appropriate, but then the title/abstract should make this clearer and broader context (HDACs in EBV and HIV latency reversal) should be acknowledged.
4) Confusing sections:
- section beginning line 299, at face value this sentence seems contradictory. If these latent proteins are expressed, but not detectable buy the immune system, then how are they logical therapeutic targets? An additional sentence or two here with example may help clarify this.
- Sentence begining line 403: if F49A-FTP treatment results in the death of latently infected cells (line 340) and CD14 and CD34 cells are the sites of latency, then shouldn't the number of CD14 and CD34 cells decrease since some are being killed?
5) Minor points: there are some awkward sentences and/or typos (lines 94, 102 (acronyms were already defined earlier), etc) but these don't hinder readability and overall this is nicely written.
Author Response
Replies to Reviewer 2
Perera and Sinclair's review on existing and new strategies to target latent HCMV is important and interesting. The author thoroughly describe several more recent studies on HDAC and I-BET inhibitors in detail, however the focus of the review is quite biased to these two approaches and lacks any discussion on several other approaches to the detriment of the overview theme proposed as the purpose of this review in both the abstract and introduction. I have attempted to present my comments in a topic-oriented approach below in order to make them more useful. In general, I believe, that with the addition of a broader literature representation, that this review would be useful and relevant to both CMV latency and the broader Viruses' audience.
1) Emphasis and audience:
- a small intro paragraph on CMV biology / replication kinetics would be more useful at the beginning (before section 2). For example, line 259 could move here.
We thank the reviewer for this idea, and have inserted a short description of the lytic replication cycle (line 25):
“The virus replicates in a broad range of cell types; briefly, after entry via fusion or receptor-mediated endocytosis, the viral genome is delivered to the nucleus where it expresses a cascade of temporally regulated genes (reviewed in [2]). This is followed by capsid assembly, the acquisition of tegument proteins, envelopment in a lipid bilayer and egress of the new virions (reviewed in [2]).”
Line 178: the literature on HIV latency reversal is extensive, and the functional overlaps here are very relevant. It would be significant (and broader impact both) to include at least a few sentences on the key HIV studies/field here for perspective and broader context.
We have added in detail on HIV shock and kill research (line 255):
“This strategy, termed ‘shock and kill,’ is also being widely investigated as a therapeutic tool to target HIV-1 latency (reviewed in [88,89]). In essence, epigenetic modifiers have been used to reactivate HIV-1 gene expression in latently infected cells to allow them to be targeted by pre-existing or newly induced HIV-1 specific immune responses. Similar shock and kill strategies against HCMV are discussed in the next section.”
We have also directed the reader to two excellent reviews of shock and kill strategies in HIV: Thomas et al., 2020 and Abner et al., 2019.
2) Missing factual references:
- lines: 57, 207, 285/287
We thank the reviewer for noticing these omissions:
For line 57 (now line 68), we have added Sinzger et al. ‘Cytomegalovirus Cell Tropism’ which details the broad tropism of HCMV and consequential multi organ disease.
For line 207 (now line 293), we have cited Krishna et al., 2017 - ‘Transient activation of Immediate Early…’
For lines 285/287 (now line 422, 424), we have cited Groves et al., 2021.
3) Content not discussed or that needs included for balance:
- current treatments also include antibody therapy. Although the title suggests focus more on antivirals, specific targeted antibodies to US28 are included, so some mention of current antibody therapy is relevant.
We thank the reviewer for this comment, and have added a section on antibody therapies (Line 171):
“2.6. Antibody therapies
A vaccine against HCMV is considered to be a very high priority, particularly for the prevention of congenital disease, but none has been licensed [63]. Due to the problems described above with respect to current antivirals, as one alternative, antibody responses have been investigated as a basis for immunotherapies [64]. For instance, hyperimmune globulin (HIG) can improve survival in patients undergoing solid organ transplantation and a range of antibodies that are capable of neutralizing the entry of cell-free virus have been developed [65,66]. However, so far, the use of neutralizing therapeutic monoclonal antibodies has only had modest effects in trials [67].”
- Section 2.3 is short and does not include the details as previous. For example what are the side effects, mechanism of action etc for letermovir (as described for the previous antivirals)?
We thank the reviewer for this comment. We have supplemented this section with (line 159):
“Adverse gastrointestinal effects have been reported with letermovir, but overall, it ap-pears to cause fewer side effects than other approved HCMV antivirals, which has been attributed to the lack of any mammalian counterpart to UL56 [58,59].”
Section 2 also doesn't discuss the newer antivirals, i.e. Maribavir, Brincidofovir, Filociclovir. Or mAbs, or Sirtuins.
We thank the reviewer for this suggestion. To ensure a more comprehensive summary of current HCMV antivirals, we have included the following (line 140):
“The main side effects of cidofovir are nephrotoxicity and myelosuppression [52]. An additional problem with cidofovir is poor bioavailability; hence a modified li-pid-conjugate of cidofovir, brincidofovir, which has improved bioavalability, is currently undergoing clinical trials for efficacy against HCMV disease [53]. Brincidofovir displays lower nephrotoxicity because unlike cidofovir, it cannot be imported by organic anion transporter 1 into epithelial cells of the renal proximal tubule [54].”
We have also added (line 163):
“2.5. Maribavir
Although not yet licensed for use, the prospective anti-HCMV drug, maribavir, has had promising results in clinical trials [60]. Maribavir competes with ATP for binding to the viral kinase pUL97. Maribavir is still active against some GCV-resistant strains of HCMV, despite the majority of GCV-resistant strains having mutations in UL97 [61]. It should be noted that maribavir could not be used in conjunction with GCV treatment as it would inhibit the initial phosphorylation and activation step of GCV [62].”
- Section 3: there are a number of other drugs that function to reactivate virus from other groups that should also be discussed. As two examples only (the others should also be included): PI3K/AKT/MEK/ERK inhibitors (Buehler, PLoS Pathogens, 2019), TNFa (Forte, mBio, 2018).
We thank the reviewer for this suggestion and have inserted the following section to make the review more complete (line 425):
“Other hypothetical drugs to shock and kill latent HCMV could also be explored. For example, it has been shown that treatment with TNFα leads to reactivation of HCMV from latency via NFκB signalling in a cell line model of CD34+ progenitor cells [109]. Fur-thermore, TNFR1 is upregulated on latently infected cells by the viral protein pUL138, which could potentially make these cells more responsive to TNFα treatment over un-infected bystander cells [110]. However, extreme caution would have to be applied in stimulating immune cells (in the case of HSCT) with the pro-inflammatory cytokine TNFα. It could exacerbate graft vs host disease and also hamper haematopoietic reconstitution [111,112]. Others have shown that MEK/ERK and PI3K/AKT signalling is important for establishing and maintaining latency [86,113,114]. Pharmacological inhibition of MEK, ERK, PI3K or AKT has been shown to result in HCMV reactivation [115]. These drugs could therefore be examined for their potential as shock and kill agents. Furthermore, Rauwel et al. showed that phosphorylation of the cellular protein, KAP1, prevented re-cruitment of repressors SETDB1 and HP1 to the MIEP during latency [96]. Drug-induced phosphorylation of KAP1 by the ATM activator, chloroquine, led to the reactivation of HCMV in naturally latent monocytes [96].”
- Page 5 includes a lot of repeats from the discussion sections of the two main referenced papers here, the author's could summarize this section more succinctly and avoid the bias to these.
We have substantially condensed the material that was previously on page 5. It now reads as follows (line 278):
“These would then be targets for HCMV-specific cytotoxic T cells (‘kill’). One biological property of HCMV that makes this type of strategy so attractive is the knowledge that in healthy seropositive virus carriers, up to 10% of circulating peripheral blood resident cytotoxic T lymphocytes (CTLs) are HCMV lytic antigen specific and potently target lytically infected cells [100].
As anticipated, treatment of both experimentally and naturally latently infected monocytes with an HDAC4 inhibitor (MC1568) did result in derepression of the MIEP and the transient induction of expression of lytic IE antigens, though the full complement of viral lytic genes was not induced by MC1568 treatment [99]. This could be construed as the most desirable outcome for a latency reversing agent for HCMV as the treatment does not result in full reactivation of the lytic viral life cycle, thereby reducing the possibility of transplanting infectious virus. Instead, latently infected cells induced to transiently ex-press e.g. IE antigens would now become a viable target for the host immune system; an extraordinary proportion (between 0.1 and 5%) of total circulating CD8+ CTLs in sero-positive donors respond to peptides derived from IE72 [101].
Indeed, when naturally latent monocytes from a seropositive donor were treated with MC1568, the donor’s autologous T cells were able to recognise and clear them, resulting in a substantial reduction in the number of cells capable of reactivating virus after differ-entiation [99]. However, these analyses were somewhat compromised by the fact that treatment with MC1568 was carried out transiently; purified latently infected monocytes were treated for 48 hours with MC1568 in the absence of other peripheral blood mono-nuclear cells which would be unlikely to lend itself to any treatment regimen in vivo.
With this in mind, Groves et al. (2021) built on this work and screened a new generation of HDACis that had lower toxicity and resulted in a substantially higher number of monocytes expressing lytic HCMV antigens in a model of a long-term treatment regime of latently infected monocytes [102]. However, in these long-term treatment studies, both old and new HDACis also resulted in full reactivation of infectious virus. Consequently, the authors also trialled inhibitors of other classes of epigenetic enzymes, such as inhibitors of BET proteins (I-BETs) for their potential as ‘latency re-versing agents’, with far more success [103].
At a large number of metazoan promoters, RNA Pol II remains ‘paused’ due to the presence of negative transcription elongation factors such as NELF and DSIF [104]. The transcription factor P-TEFb (comprising CDK9 and Cyclin T1) can overcome this promoter pausing and allow elongation of the nascent RNA transcript to continue. However, PTEFb may be bound to the inactivating 7SK snRNP or complexed to the BET protein, Brd4, reducing levels of free PTEFb. Treatment with I-BETs can release more PTEFb, resulting in increased transcription from Brd4-independent genes [105,106].”
Section 4: targeting during transplant. There is extensive work using HCMV homologs in animal models for transplant that should be acknowledged, i.e. MCMV renal transplant (Abecassis lab) and RCMV heart transplant (Streblow lab). There is also evidence for ex vivo treatments in other scenarios (outside if CMV) that could be referenced to provide proof of principle to this approach (i.e. Ku, 2020 for EBV; Galasso, Nat. Comm, 2019 lung transplant). -- If the desired focus of this review is to emphasize HDAC inhibitors for example rather than general latency reversal targets, this would be appropriate, but then the title/abstract should make this clearer and broader context (HDACs in EBV and HIV latency reversal) should be acknowledged.
We thank the reviewer for this suggestion and have added more details as follows (line 541):
“The possibility of targeting viral pathogens in donor organs prior to engraftment is not without precedent. Ex-vivo delivery of monoclonal antibody to target EBV as well as light-based therapies to target HCV in donor lungs prior to transplant have been de-scribed ([126,127]. Similarly, removal of macrophages from RCMV-infected donor hearts prior to transplant reduces chronic rejection ([128].”
4) Confusing sections:
- section beginning line 299, at face value this sentence seems contradictory. If these latent proteins are expressed, but not detectable buy the immune system, then how are they logical therapeutic targets? An additional sentence or two here with example may help clarify this.
We thank the reviewer for this comment, and have inserted the following (line 451) to highlight that the virus has evolved alongside the immune system, enabling it to conceal certain antigens, but this does not apply to drugs.
“However, the latent virus also expresses protein coding genes which should theoretically be targetable by the immune system, but these antigens still manage to avoid detection via a number of latency-associated immune evasion strategies employed by HCMV (reviewed in [26]).”
- Sentence begining line 403: if F49A-FTP treatment results in the death of latently infected cells (line 340) and CD14 and CD34 cells are the sites of latency, then shouldn't the number of CD14 and CD34 cells decrease since some are being killed?
We have amended this paragraph with this clarifying statement (line 572):
“Only a very small proportion of CD14+ and CD34+ cells are naturally latently infected, so a significant decrease in the total number of these cells would not be expected [133].”
5) Minor points: there are some awkward sentences and/or typos (lines 94, 102 (acronyms were already defined earlier), etc) but these don't hinder readability and overall this is nicely written.
We thank the reviewer for this comment. Line 112, 113: these redefinitions have been removed.

Reviewer 3 Report
This is a review article from Perera and Sinclair, entitled, “HCMV Antivirals and Strategies to Target the Latent Reservoir”. This is quite well-written, and it provides a very comprehensive dive into the literature on the topic. Very minor comments are provided for the authors’ consideration, which are suggestions to merely complement an already strong review of the literature. These are as follows:
-Consider adding a brief paragraph on maribavir. Although this anti-viral is not currently approved, it has Orphan Drug designation (EU) and Breakthrough Therapy designation (FDA). Might be worth mentioning.
-Very, very minor - Consider combining several, short paragraphs in a few places into one larger paragraph for improved readability. The 3 sections are: A) lines 214-231 into 1 paragraph; B)lines 253-276 into 1 paragraph; C) lines 277-288 into 1 paragraph.
Reviewer 4 Report
The manuscript is very well written. The topic is highly relevant. Very important aspects are discussed. With very few exceptions (see below), the recent literature is covered.
- I understand that the authors just briefly introduce the drugs that are clinically approved, however, Brincidofovir (BCV) (and maybe Maribavir?) might be considered to be included (https://www.sciencedirect.com/science/article/pii/S0166354218306570?via%3Dihub; https://pubmed.ncbi.nlm.nih.gov/31532960/).
- Why are FOS and CDV combined in one paragraph?
- Line 123: I am not sure that the statement concerning the uncharged nature of CDV is true - at least in vivo. I would have thought that the phosphate group would be anionic. Accordingly, CDV (but not BCV) is transported by renal organic anion transporters such as hOAT1, and CDV needs to be administered together with Probenecid (https://pubmed.ncbi.nlm.nih.gov/17372702/; https://pubmed.ncbi.nlm.nih.gov/10703662/; https://www.ncbi.nlm.nih.gov/pmc/articles/PMC5113238/; https://www.gilead.com/~/media/Files/pdfs/medicines/other/vistide/vistide.pdf). In any case, the reader is a bit puzzled by the statement that CDV being a monophosphate is stated to diffuse freely, while the authors argue that GCV acquires a charge through phosphorylation which retains GCV in cells (line 85).
- Given the data published by authors including Skip and Luka concerning effects of interferons (IFN) on CMV latency (https://pubmed.ncbi.nlm.nih.gov/24586165/; https://pubmed.ncbi.nlm.nih.gov/9687534/) and the availability of clinically approved recombinant interferons (https://www.drugs.com/drug-class/interferons.html), I feel that this aspect might be worthwhile to be covered. It might even be put into perspective to shock-and-kill approaches (S&K) since the HCMV-MIEP seems to be IFN responsive (https://pubmed.ncbi.nlm.nih.gov/15795289/).
- Based on the expression of pUL138 during latency, the finding that pUL138 enhances TNFR1 surface disposition (https://pubmed.ncbi.nlm.nih.gov/21880774/, https://pubmed.ncbi.nlm.nih.gov/21976655/, https://pubmed.ncbi.nlm.nih.gov/23580527/), the relevance of NF-kB signalling for HIV in the context of S&K (see e.g., https://journals.plos.org/plospathogens/article?id=10.1371/journal.ppat.1005066), the NF-kB responsiveness of the HCMV MIEP (e.g., https://pubmed.ncbi.nlm.nih.gov/11831701/; https://pubmed.ncbi.nlm.nih.gov/11522776/), the levels of TNF/IL1b in GvHD Tx patients and longstanding discussions concerning the role of TNF in HCMV reactivation (e.g., https://pubmed.ncbi.nlm.nih.gov/7905100/), I think that this additional facet would fit very well to the current review.
- Although the paper is cited (Ref. 69), I missed a slightly longer discussion of the druggable KAP1/Trim28 aspect of HCMV latency (https://pubmed.ncbi.nlm.nih.gov/25846574/).
Round 2
Reviewer 2 Report
I appreciate the authors' thoughtful responses to my (and the other reviewers') comments. The manuscript is sound and much improved and I feel it is a valuable contribution to the field.